# In Vitro Evaluation of Structural Factors Favouring Bacterial Adhesion on Orthodontic Adhesive Resins

**DOI:** 10.3390/ma14102485

**Published:** 2021-05-11

**Authors:** Roberta Condò, Gianluca Mampieri, Guido Pasquantonio, Aldo Giancotti, Paola Pirelli, Maria Elena Cataldi, Serena La Rocca, Andrea Leggeri, Andrea Notargiacomo, Luca Maiolo, Patrizia De Filippis, Loredana Cerroni

**Affiliations:** 1Department of Clinical Sciences and Translational Medicine, University of Rome “Tor Vergata”, Via Montpellier, 1, 00133 Rome, Italy; gianluca.mampieri@uniroma2.it (G.M.); guido.pasquantonio@uniroma2.it (G.P.); giancotti@med.uniroma2.it (A.G.); p.pirelli@gmail.com (P.P.); sere.larocca@gmail.com (S.L.R.); cerroni@uniroma2.it (L.C.); 2PhD in Materials for Health, Environment and Energy, University of Rome “Tor Vergata”, Via della Ricerca Scientifica, 1, 00133 Rome, Italy; melena.88@hotmail.it; 3Oral Surgery Specialty School, University of Rome “Tor Vergata”, Via Montpellier, 1, 00133 Rome, Italy; dr.andrealeggeri@gmail.com; 4Institute for Photonics and Nanotechnologies-National Research Council, Unit of Rome, Via Cineto Romano, 42, 00156 Rome, Italy; andrea.notargiacomo@ifn.cnr.it; 5Institute for Microelectronics and Microsystems-National Research Council, Unit of Rome, Via del Fosso del Cavaliere, 100, 00133 Rome, Italy; luca.maiolo@cnr.it; 6Department of Biomedicine and Prevention, University of Rome “Tor Vergata”, Via Montpellier, 1, 00133 Rome, Italy; patrizia.de.filippis@uniroma2.it

**Keywords:** orthodontic adhesive resin, bacteria adhesion, surface roughness test, FIB/SEM analysis

## Abstract

Bacterial adhesion to the surface of orthodontic materials is an important step in the formation and proliferation of plaque bacteria, which is responsible for enamel demineralization and periodontium pathologies. With the intent of investigating if adhesive resins used for bracket bonding are prone to bacteria colonization, the surface roughness of these materials has been analyzed, combining information with a novel methodology to observe the internal structures of orthodontic composites. Scanning electron microscopy, combined with focus ion bean micromachining and stylus profilometry analyses, were performed to evaluate the compositional factors that can influence specific pivotal properties facilitating the adhesion of bacteria to the surface, such as surface roughness and robustness of three orthodontic adhesive composite resins. To confirm these findings, contact angle measurements and bacteria incubation on resin slide have been performed, evaluating similarities and differences in the final achievement. In particular, the morphological features that determine an increase in the resins surface wettability and influence the bacterial adhesion are the subject of speculation. Finally, the focused ion beam technique has been proposed as a valuable tool to combine information coming from surface roughness with specific the internal structures of the polymers.

## 1. Introduction

Enamel demineralization induced by organic acids, is commonly recognized as one of the orthodontic treatment complication, as it occurs in about 50% of patients undergoing fixed therapy [1,2].

Indeed, oral application of orthodontic appliances usually determines the increase of bacterial population and the change in oral microbiota with a growth in oral pathogens, as periodontopathic Gram-negative bacteria and cariogenic streptococci [3]. *Aggregatibacter actinomycetemcomitans* (AA) play an important role in activating the immune-inflammatory response in gingival tissues [4]. *Mutans streptococci* (MS) proliferation in dental plaque precipitates an increase in cariogenic effects. In any case, these levels of bacteria return to normal after a complete removal of the orthodontic device [5,6,7,8].

B.S. Lim reported that the orthodontic adhesives have a greater ability to retain cariogenic streptococci in the material, which constitutes the bracket [9]. In this perspective, orthodontic adhesives might have more favourable characteristics for bacterial adhesion than bracket materials [10].

A. Gwinnett noted that, one of the most common sites where demineralization occurs is the junction of adhesive resin and enamel [11].

To this end, bacterial adhesion to the surface of the adhesive is an important step in the formation of plaque and enamel demineralization.

Moreover, R. Weitman claimed that, the predisposing factor to demineralization lesions is the adhesive remaining on the enamel surface, right around the base of the bracket, since its rough surface can lead to a rapid adhesion and growth of oral micro-organisms [12].

Indeed, rough surfaces tend to harbour bacteria, promote pigment absorption and increase surface deterioration [13].

A rough surface, promoting bacterial colonization, increase in fact also the area of bacterial adherence [14].

Orthodontic adhesives differ from each other not only in the chemical composition of the polymeric matrix, for setting mechanisms and reactivity of the surface-level constituents, but also in their surface features. As with real objects, orthodontic adhesives are usually quite rough displaying surface irregularities at the micro scale.

Surface roughness (SR) and surface free energy (SFE) are considered key elements in determining the demineralization of enamel when the excess adhesive material around the bracket favours the bacterial plaque accumulation [15,16,17].

According to findings reported in S. Ahn et al., the analysis of four orthodontic adhesive systems by confocal laser microscopy and tensiometer showed no statistically significant correlations between SR and adhesion of MS, although the SR values were different in all adhesive systems, thus suggesting that initial adherence of MS is influenced by SFE characteristics of the adhesives rather than SR [18]. Conversely, M. Quirynen recorded significant differences in the SFE values of different polymeric materials and observed correlations between these values and bacterial plaque adhesion, suggesting that SR may not be the most important factor in affecting the adhesion of the MS to the surface of the adhesive material. The differences may be related instead to the chemical nature of the adhesive [15].

In particular, chemical–physical interactions, such as van der Waals forces, or acid–base interactions, play an important role in the initial bacterial adherence and can be defined by SFE and its components [14].

The filling material contained in the formulation of the adhesive composite resin is, for a good percentage, constituted by inert glass, which seems to contribute to a lower SFE and polarity [19].

Since the higher the SFE, the greater the amount of bacteria, the greater adhesion of SM to resin-modified glass ionomer cements compared to composite resins can be explained by the respective SFEs of their surface [16].

Although it is difficult to establish whether, SFE predominates over SR (or vice versa) with regard to bacterial adhesion, there is scientific evidence of interaction between the two surface characteristics [20].

Thus, the clinical importance of the influence of SR on plaque formation justifies the need for smooth surfaces with a low SFE even with regard to adhesive resins. It is essential to establish the bacterial adhesion pattern to orthodontic adhesives, to prevent enamel demineralization.

As reported in the literature, physical properties such as surface roughness, hydrophobicity, and the composition of material play an important role in bacterial adhesion. Although, this can be considered as true also for orthodontic composite resins to date, in scientific literature there are still too few studies on the argue [9,10,16,18,21,22]. Especially in case of bracket bonding, where the resins have not subjected to finishing and polishing procedures.

Therefore, the aim of this in vitro study was to investigate the peculiar, over mentioned, physical properties in three different light-cured orthodontic adhesive resins used for bracket bonding in order to provide more and valuable information on bacterial adhesion. Furthermore, in this work, we proposed the focused ion beam (FIB) technique as a valuable tool to compare the internal structures of the polymers, thus correlating the material composition to the specific surface roughness of the resin and to the probable more favourable adhesion of bacteria. Additionally, the results are correlated with wettability measurements and with bacterial adhesion tests to provide a complete picture of the phenomena occurring at adhesive level.

## 2. Materials and Methods

Three different light-cured orthodontic adhesive resins were considered in this study Table 1. 

## 3. Sample Preparation

Specimens of each adhesive material to be tested were obtained using thermo-formed polyurethane moulds, reproducing the negative shape of a disc. Three moulds have been completely filled with each of the three orthodontic resins considered in the study and, before the polymerization process, a transparent strip (Hawe Neos Dental, Bioggio, Switzerland) has been placed on and pressed to create a smoother and more uniform surface as possible. Glass plates were placed on the top and bottom of the mould to provide flat surfaces.

A light-curing unit (LCU) (LED starlight lamp) has been used, whose power density was previously measured with a curing radiometer (Model 100, Demetron Research Corp., Danbury, CT, USA, Serial no. 129540) and then set at 400 mW/cm^2^.

The orthodontic adhesive resins have been light-activated as recommended by each manufacturer, applying the LCU at the top and bottom surfaces, where the light tip was placed in contact with the glass plate at a distance of 1.0 mm from the specimens. 

Eight test discs (10.0 mm in diameter and 4.0 mm in thick), made of each orthodontic resin in exam, have been obtained for a total of 24 specimens. In order to reproduce a situation as close as possible to the clinical conditions, no finishing and polishing were carried out. Orthodontic composite resins are used exclusively in bracket bonding procedures as a junction material that is interposed between the dental enamel and the base of the bracket. The only surfaces exposed to the hostile environment of the oral cavity are the surrounding ones, which are however minimal. For these reasons, they are inaccessible to such mechanical surface finishing and polishing manoeuvres and therefore not foreseen in the clinical operating protocols.

### 3.1. Scanning Electron Microscopy (SEM) Characterization

Before characterization, all samples have been cleaned with acetone to eliminate impurities, rinsed in deionized water, dehydrated, fixed on stubs and coated with a thin (few nanometres thick) gold film obtaining by sputtering. This last step was made using Bio-Rad SEM (Microscience Division, Hemel Hempstead, UK) Coating System. 

Samples were divided into three randomly selected groups and stored in glass containers with distilled water at 37 °C until they have been used for surface morphological investigations through SEM imaging with magnification from 5000 to 60,000×, obtained by Leo Supra 35 FE-SEM Field Emission Scanning Electron Microscope (Carl Zeiss Smt, Oberkochen, Germany).

### 3.2. Cross Sectional Focus Ion Bean/Scanning Electron Microscopy (FIB/SEM) Analysis

FIB milling was utilized to study the inner structure of the materials and to evaluate the use of this technique to obtain a qualitative assessment of the materials robustness. FIB processing employed a FEI-Helios Nanolab 600 (FEI/Thermo Fisher Scientific Phillips, Research Center of Eindhoven, Eindhoven, The Netherlands) dual-beam equipment with a Ga^+^ ion source operated at 30 keV acceleration energy and 6.5 nA ion current. By using the built-in pattern generator, rectangular areas were treated at a fixed dwell time of 1 µs and ion dose values in the range 5–15 nC/µm^2^. The field-emission SEM present in the same dual-beam vacuum chamber was employed for imaging using an acceleration voltage of 5 kV and a beam current of 680 pA. Both FIB milling and SEM imaging have been performed at room temperature. To reduce drifts due to sample charging, a 200 Å thick chromium film has been deposited by evaporation using a shadow mask. Silver paint was then put on the sample surface to properly connect the chromium layer to the grounded holder. Electron imaging using scanning ions as probe was also performed to evidence differences in crystalline structure and density of the polymeric features thanks to the grain orientation contrast provided by this technique. Working at various tiled angles, the milling procedure was implemented to investigate both structural composition than samples hardness.

### 3.3. Surface Roughness Test

The opposing bases of the two hollow cylinders of a micrometer were first coated with an insulator material (Isopraim, Perident Dental Product s.r.l., Florence, Italy) and then used as a mould to make 12 discs of 2 mm in height and 6 mm in diameter in each of the 3 orthodontic composite resins taken in exam Table 1. The wheel of the instrument was turned, which allows the two cylinders to move closer together, until the micrometer measured 2 mm. Any excess material was manually removed with a Heidemann spatula and the sample was made uniform by means of a transparent matrix. Each disk underwent to polymerization by a VALO curing light (Ultradent, UT, USA), according to the times indicated by the manufacturer. The 36 samples thus obtained were subjected to surface roughness test using Surftest SJ-210 Portable Surface Roughness Tester (Mitutoyo, Kawasaki City, Japan). 

The instrument complied with ISO 4287: 1997 (Geometrical Product Specifications (GPS)-Surface texture: profile method-terms, definitions, and surface texture parameters) [23].

The roughness data were obtained by recording the vertical movements of a stylus (with a radius of 2 µm and an angle of 60°) sliding across the sample.

The parameters set for roughens evaluation were:

The set parameters set for roughens evaluation were: 

cut off length: λc = 0.8 mm; λa = 2.5 μm; 

measurement force = 1 N; 

measurement speed = 0.25 mm/s; 

range: AUTO.

Each sample was inserted on the appropriate support. The stylus tip was, positioned in the center of the upper face of the disc before initiating surface roughness scan. 

For each sample of resin, 12 measurements were performed, allowing to evaluate the following parameters: 

R_a_—average roughness: the arithmetic average of the absolute values of the roughness profile ordinates; 

R_q_—root-mean-square roughness: the root mean square along the sampling length;

R_z_—average depth: the average depth maximum peak to valley of five consecutive sampling lengths within the measuring length.

The collected data were subjected to statistical analysis.

### 3.4. Contact Angle Measurements

To evaluate the wettability of the sample surface for the different materials, contact angle measurements have been performed. Three specimens for each material have been prepared, without surface polishing to resemble the clinical conditions. The measurements have been carried out at 25 °C by depositing on each slice ten drops of filtered and distilled water with a volume of about 500 μL. Images have been captured by a Reflex Nikon D810 (Nikon Corporation, Tokyo, Japan) with a Nikon 60 mm f/2.8 micro and contact angle have been analyzed and averaged on the 10 different measures. 

### 3.5. Bacterial Adhesion Testing

*Aggregatibacter actinomycetemcomitans* ATCC 43718 has been useed to indirectly measure the adhesion properties of bacteria onto the proposed resins. A lyophilized strain was reconstituted with Brain Heart Infusion Broth implemented as medium, maintaining the temperature at 34 °C for an incubation time of 48 h in a modified atmosphere (5% CO_2_). Then, the cultured broth (2 mL) was extracted and inoculated in a vial, adding further medium, reaching a total volume of 5 mL. For each material we adopted a positive control, inserting the resin disk in the vial using another one as reference. The vials have been incubated for 24 h adopting the same conditions (34 °C and modified atmosphere).

We plated 9 mm-well by using Brain Heart Infusion Agar and a 0.5 mL volume of bacterial suspansion extracted from both the vials (positive and negative control). To better quantify the bacterial colony different concentrations have been investigated (10^−4^, 10^−3^, 10^−2^). These wells have been incubated at previous conditions and monitored every 24 h. Finally, solution samples have been prepared and assorbance measurements have been collected by using a UV–vis spectrophotometer (Bausch & Lomb, New York, NY, USA) at a wavelength of 580 nm to calculate the concentrations of bacteria per mL.

### 3.6. Statistical Analysis

Data were analysed using Kruskal–Wallis and Mann–Whitney U tests; Bonferroni Scheffe, and Sidak multiple comparison tests were used, *p* values were computed and compared with statistical significance at the *p* = 0.05 level. The data were analyzed with the statistical software STATA (STATA Statistical Software release 12.1; Stata Corporation, College Station, TX, USA).

## 4. Results

### 4.1. SEM Results

Figure 1, showed the tilted (52°) views of the sample surfaces obtained by SEM at a magnification of 20,000×. It was apparent that the surface structures of Bisco (a) and Leone (b) resins were quite similar, showing a grainy like morphology with micron scale grains. Conversely, the TXT resin surface (c) appeared blunter.

### 4.2. Cross Sectional FIB/SEM Results

FIB milling experiments were performed on the three different materials by scanning the FIB on 13 × 16 µm wide rectangular regions at a dose of 7.5 nC/µm^2^. The milling process time was 180 s. After a very fast step in which the metallic coating of chromium is removed (about 2 s) as effect of the ion impact, the resin material is sputtered away producing a rectangular hollow with an irregular bottom surface. As visible in Figure 2, the resulting average depth is strongly sample dependent, with an average milling rate that is almost 2× and 1.7× that of TXT and Leone resins, respectively.

Morphological characteristics and differences of the samples structure have been investigated in more detail by cross-sectional SEM analysis performed on the vertical sidewalls of the hollows created by FIB milling. SEM images, collected at a magnification of 60,000× are shown in Figure 3. Very large grains with light grey/white contrast are present in a matrix made of a main phase displaying dark contrast in the images and small features having size in the micro/sub-micro scale.

These findings are more clearly evidenced in the panels of Figure 4. Figure 4a shows the image of the bottom surface of TXT resin obtained using the ion microscopy which is more effective in highlighting the different phases (see Figure 3c for comparison), and thus the grain boundaries, due to the different interaction of the beam with the materials. Indeed, ions are heavier and can produce a superior contrast in imaging different materials. More specifically, the very large grains now display a black contrast. As for the vertical sidewalls of the hollows produce by FIB milling, a finer polishing was made using FIB treatment at low ion current (i.e., below 100 pA) producing smoother surfaces. In this condition the different features and phases are better defined especially those having nanoscale size. The cross-sectional view of the Bisco resins reported in Figure 4b shows the presence of a very inhomogeneous crystalline structure (medium and large size features identified in Figure 4b with two and three arrows, respectively) embedded in a more homogenous matrix made by small grain with size of hundreds of nm (single arrow).

### 4.3. Surface Test Results

Table 2, Table 3 and Table 4 show the values relating to surface roughness of the orthodontic adhesive resins under examination. 

The collected data were subjected to statistical analysis, which allowed to obtain average measurements of R_a_ equal to: 0.472 ± 0.176 µm for TXT, 0.539 ± 0.219 µm for Bisco and 0.543 ± 0.232 µm for Leone. 

The R_a_ means of the three orthodontic adhesive resins were then inserted into a comparison graph (Figure 5).

### 4.4. Contact Angle Measurements

All the materials exhibited a similar behavior with contact angles ranging from 60° to 65° (see Figure 6). No notable variations have been observed in wettability, suggesting that surface roughness does not differ significantly from one resin to another as also confirmed by the SR measurements. 

In these conditions, all the three resins showed a more hydrophilic behavior, justifying the hypothesis that the presence of liquid in the exposed part of the resin between the tooth and the bracket can favor bacteria colonization.

### 4.5. Bacterial Adhesion Testing

As can be seen in Table 5, no significant variations have been observed in bacterial concentrations for the three different resins. This indirect measurement perfectly matches the findings collected with contact angle tests and surface roughness of the samples where a similar adhesion behavior is expected. We believe that this preliminary result can be sufficient to confirm the assumption that surface roughness is a crucial indicator for a material in the capacity of offering a favourable environment for bacterial colonization.

## 5. Discussion

The analyses performed in this in vitro study allowed us to reveal the structural nature of the three orthodontic adhesive resins and, above all, to understand how this structure influences the properties of these resinous polymers, such as the surface roughness and robustness. The information on contact angle and bacteria proliferation have been added to demonstrate clearly that in these specific materials the surface roughness is a valuable factor to attribute bacteria adhesion on that resins. Moreover, the FIB analysis has been used to correlate internal structure of the polymers with the specific surface roughness, thus explaining the little differences among the materials.

The SEM investigation reported in Figure 1 shows that the sample morphology and the surface structures of Bisco (Figure 1a) and Leone (Figure 1b) are quite similar.

A numerous series of irregular grains, with a grain size of the order of a few hundred nanometers, were inhomogeneous in shape and distribution and appeared incorporated in a matrix of a very similar shade of grey, more intense in the Bisco adhesive in which, moreover, the crystalline irregularities were sharper and better defined.

TXT resin surface (Figure 1c) appeared homogeneous and blunt instead: on a lighter grey polymeric matrix, there were sparse crystalline grains, of small, regular, and roundish shape which were sometimes interspersed randomly around depressed points or areas. 

These differences are even better observed at a greater magnification in the cross-sectional SEM images (Figure 3) in which it is possible to understand that the grey tone, more or less intense, of the organic matrix of the Bisco (a) and Leone (b) resins is due to the presence in it of numerous and small roundish crystalline granules, which are uniformly distributed, probably mixed with it, so as to create a single granular mass in which there are also single white crystals of various irregular and polyhedral shapes of intermediate and large dimensions. The TXT adhesive (c) has instead revealed a different structure where clear and clean rare very small granules, and some crystals, are randomly distributed in a homogeneous matrix of intense black. These granules are larger in size respect to the other polymers, from medium to large, with a clear polyhedral morphology and a bright white color.

As visible in Figure 2, after FIB milling experiments, the resulting average depth is strongly sample dependent. More specifically, the average milling rate of Bisco is almost 2× that of TXT and 1.7× that of Leone. 

The SEM images of the sample surfaces after FIB milling show the appearance of peculiar residual cavities that allowed to measure in situ not only the depth of the engraved volume and therefore the robustness of each orthodontic composite resin but also to observe the different internal structures characterized by at least two solid phases. 

Moreover, the milled area is not homogenous, i.e., a rough surface is generally obtained after FIB milling, and apparent monolithic protruding features are present showing a characteristic contrast, as in the case in Leone and TXT samples. 

Particularly, the milled surface of Bisco samples exhibited a less smooth surface with the presence of filamentary features likely due to a ‘plastic’ rearrangement of redeposited material.

A composition of two different material patterns can be easily recognized in all the samples.

As an example, the scanning ion microscopy image of TXT material reported in Figure 4a shows features with a sharp black contrast level on a smooth grey background: these features are clearly corresponding to the protruding structures visible in Figure 2c, which are less sensitive to the ion milling respect to the polymeric matrix. 

Moreover, the FIB/SEM cross sectional analysis reveals a similar inhomogeneous vertical structure in all the three samples (see Figure 4b for the case of Bisco sample) with medium and large size features (bright contrast) embedded in a homogenous matrix in which small grains measuring hundreds of nm can be recognized.

Through surface test, it was possible to compare the surface structure of the three orthodontic composite resins with each other.

From a first comparative analysis it is evident that the parameters R_a_, R_z_, and R_q_ are similar in all three materials; specifically, an almost identical average is observed for Bisco and Leone resins, concerning all three parameters taken into consideration Table 2 and Table 3. Same results have been observed on contact angle measurements, thus suggesting that all the three resins show a more hydrophilic behavior and likely the same clinical performance in term of possible bacteria adhesion.

On the other hand, the average values recorded for TXT are slightly different Table 4 from the previous ones (Bisco and Leone). 

In literature, these minor changes can be explained by the internal composition of each resin, depending on various factors such as the size of the filler, the percentage of filler particles, the hardness, and the degree of conversion of the polymer itself [24].

It has also been claimed that high R_a_ values are associated with materials with large filler particles and irregular in shapes [25].

Therefore, considering this information and transferring it to our in vitro study, we can infer that the Bisco and Leone resins, with superimposable surface test values and higher than those shown by TXT, have a rather similar structural and probably also chemical composition.

The similarities resulted in Bisco and Leone structures observed in SEM analysis, are also confirmed by a chemical composition declared by the manufacturers, which appear very similar in the two adhesive resins Table 1. In fact, TEGDMA and UDMA appear as components of the organic matrix in both Bisco and Leone, which includes Bis-GMA, also. In particular, UDMA is completely absent in TXT, which is instead an orthodontic polymer based on TEGDMA, BisGMA and bis-EMA loaded with about 70–80% in crystalline fillers. 

S.D. Murray argued that it is the high percentage of fillers that provides adequate bond strength [26]. A.E. Papakonstantinou stated that adhesives formulated with UDMA monomers produce resins with viscosities comparable to Bis-GMA as in the case of TXT. Adhesives formulated with a high percentage of UDMA can be used to produce resins with higher viscosity and higher bond strength, potentially without affecting their degree of conversion [27].

From the safety data sheets in Table 1, it is reported that TXT mainly contains in its chemical composition reaction products with quartz, as fillers. In its crystalline form, quartz, hard and chemically inert, is usually used as a macro-filler (particles of 8–120 µm). The resins Bisco and Leone, on the other hand, encloses essentially silica, which in the pyrogenic form is in the shape of small spheres and is used as a filler with 0.04 µm micro-particles.

It has been observed that, the more regular and smaller the loading particles of the material, the greater the possibility of obtaining surface smoothness [28].

Despite the larger size of the fillers in TXT, the greater dispersion of the quartz particles in the polymeric matrix could be the reason at the basis of the lower surface roughness value obtained about it, after surface test.

R_a_ values of less than 1 μm provide the material a visually smooth appearance, because of its wavelength which is larger than that of the visible light [28].

However, roughness of 0.2 μm is considered as the initial limit for bacterial accumulation [29,30,31].

The purpose of four recently published studies was to search for a correlation between the surface characteristics of the resins (or other orthodontic materials under examination) and the adhesion of cariogenic streptococci [9,10,18,32].

These studies show that the mean roughness of the TXT, calculated by confocal laser scanning microscopy, was found to be 0.39 ± 0.02 μm [8]. It is important to underline that this value is very close to the result obtained in our in vitro study in which TXT was also characterized by having the lowest surface roughness value compared to Leone and Bisco resins. All the R_a_ values obtained from the surface test on the three orthodontic adhesive resins have, on average, exceeded the initial limit of bacterial accumulation, on the other hand, N. Beyth reported that if the R_a_ values exceed this limit, in addition to a greater accumulation of plaque, such uneven surfaces can acquire retention niches for bacteria and act as shelters [33].

Furthermore, it has been observed that the higher the roughness of the artificial material surface, the more complicated its cleaning will be compared to materials with smoother surfaces [34,35,36,37]. In the literature, it has also been reported that, surface roughness (SR) of biomaterials is able to influence biofilm formation and that changing in surface properties related to the orthodontic bonding procedure may also meaningfully affect the bacterial biofilm proliferation just around the orthodontic appliances [4,38].

In fact, the development of a dysbiotic oral microbiota and the related growth of the biofilm create problems both for the natural tissues, hard or soft, of the oral cavity and for the artificial biomaterials inserted [39].

In light of our results obtained from the structural morphology evaluations and surface roughness measurements, we therefore believe that the three orthodontic adhesive polymers tested in our study can be all considered as possible bacteria receptacle materials.

Also, the FIB measurements confirm the presence of inhomogeneous crystalline structures embedded in the polymer matrix for the three resins. These findings justify the peculiar shape of the surface roughness of the examined samples. Indeed, the FIB analysis can reveal the internal structure of these polymers and can explain the specific chemical and mechanical resistance to bacteria and acids giving information also on how the surface roughness can interact with bacteria (i.e., a different level of porosity). FIB milling used in the present in vitro study to investigate the internal structure of three orthodontic adhesive resins, allowed us to make some considerations about the different size and type of the crystalline granules dispersed into the polymeric matrix and the capability of resistance offered by each polymeric material to the ionic milling. The first information can justify the specific type of surface of the resin while the experimental data reported on the FIB milling speed allowed to acquire new information regarding the single ability demonstrated by the three adhesive resinous polymers to offer more or less resistance to scratching. Although it is not possible to directly correlate hardness measurements of materials, at least the milling depth can be defined as a valuable method to validate the internal chemical and physical structure of the different adhesive resins. Until now FIB, as a destructive analysis, has been used mainly as a material preparation technique to create transmission electron microscopy (TEM) lamellae. Recently, M. Sezen has performed micro and nanostructural analysis of a human tooth using correlated FIB and TEM investigations to reveal different morphological characteristics of dental tissue via complementary imaging and diffraction analysis [40]. In light of this finding, we believe FIB can be also implemented as a comparative technique to evaluate the internal structure of different polymers.

To this end, the investigations carried out in this study allow us to hypothesize the use of the focused ion beam as a possible technique for marking the hardness of dental biomaterials, in general. However, this hypothesis will have to be carefully corroborated, providing a systematic comparison with the hardness parameters obtainable with other commonly used techniques, such as Vickers’ hardness test. It would be worthwhile to be able to deepen this technique, to establish the resistance values to ion milling or scratching among the various procedures of physical characterization of dental biomaterials.

In our opinion, also FIB imaging can represent a complementary tool to better understand physical and chemical properties of orthodontic materials to be combined with mechanical tests and SEM microscopy.

These findings have been confirmed by contact angle measurements and investigating the bacterial adhesion testing where no significant variations have been observed among the three resins, suggesting that the surface properties can give a valuable indication of the tendency of these materials to be colonized by bacteria during the clinical treatments.

## 6. Conclusions

Surface roughness measurements were found to be quite similar between the three orthodontic adhesive resins examined. Despite this, the SR analysis revealed a greater morphological similarity in the structural organization of Bisco and Leone, the two resins with similar chemical composition. Their Ra parameters showed slightly higher values than those detected in TXT, despite the latter being made up of larger fillers, as observed in the Fib/SEM images. No noticeable variation in the wettability proves that the surface roughness does not differ significantly from one resin to another, as confirmed by the SR measurements. All resins show a fairly hydrophilic behavior, thus justifying a possible bacterial colonization of their surfaces. Indirect tests on bacteria adhesion confirm similar trends for all the three materials, as expected after the previous investigations. It is therefore essential that the orthodontist always carefully removes excess orthodontic adhesive resin that extends beyond the base of the attachment and which can become a dangerous receptacle for cariogenic and periodontopathic bacterial species, which always protects all neighboring areas of dental enamel with fluorinated varnishes and performs, as far as possible, the finishing and polishing of the residual resin.

## Figures and Tables

**Figure 1 materials-14-02485-f001:**
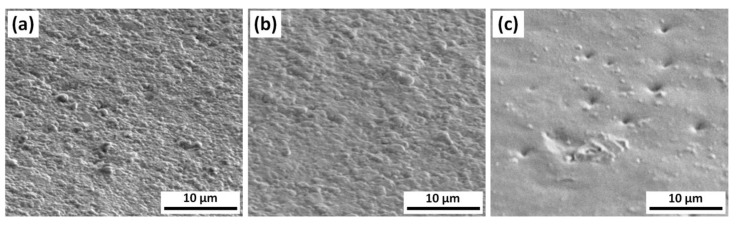
SEM images of the surfaces of (**a**) the Bisco Ortho Bracket Paste LC (Bisco), (**b**) Light-Cure Orthodontic Paste (Leone), and (**c**) Transbond XT^TM^ Light Cure Adhesive (TXT).

**Figure 2 materials-14-02485-f002:**
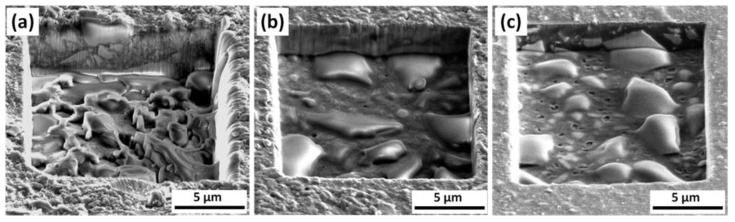
Morphological investigation of the resins after FIB milling tests. (**a**) for the Bisco Ortho Bracket Paste LC (Bisco), the milling rate is almost double respect the other two samples. At the center of the milled region is possible to observe redeposited material sputtered from the side to the middle of the squared hole. (**b**) Light-Cure Orthodontic Paste (Leone). (**c**) Transbond XT Light Cure Adhesive (TXT) after FIB milling process.

**Figure 3 materials-14-02485-f003:**
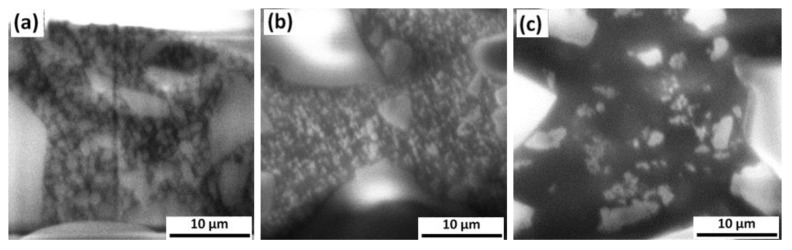
Cross-sectional analysis: SEM images collected on the sidewalls of the FIB milled areas on (**a**) the Bisco Ortho Bracket Paste LC (Bisco), (**b**) Light-Cure Orthodontic Paste (Leone), and (**c**) Transbond XTTM Light Cure Adhesive (TXT) after FIB milling.

**Figure 4 materials-14-02485-f004:**
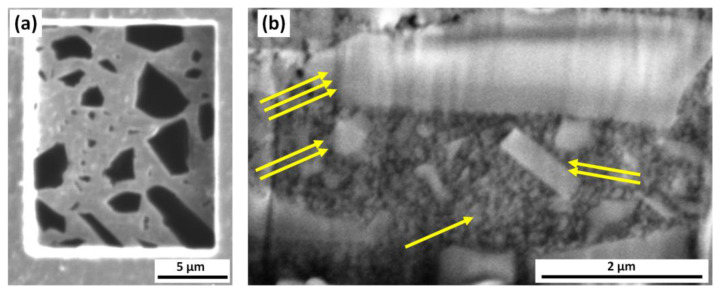
(**a**) Ion microscopy image showing two well contrasted phases in the bottom surface of the milled area of Transbond XT^TM^ Light Cure Adhesive (TXT); (**b**) detail of the polished cross section of Bisco Ortho Bracket Paste LC (Bisco).

**Figure 5 materials-14-02485-f005:**
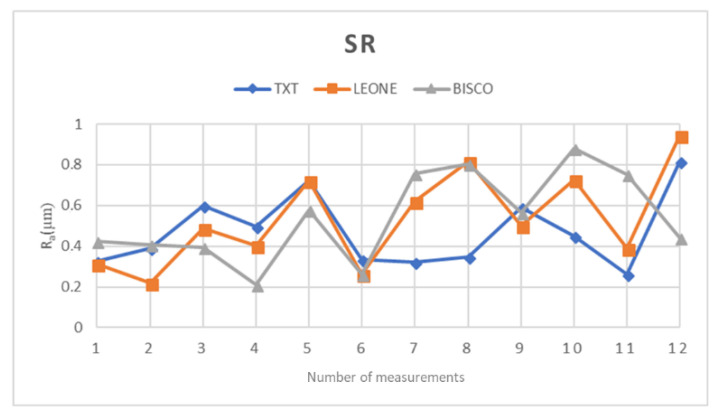
Graph illustrating the R_a_ means of the three examined orthodontic adhesive resins.

**Figure 6 materials-14-02485-f006:**
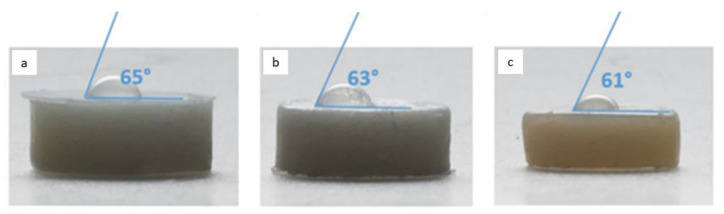
Values of the contact angles measured on (**a**) the Bisco Ortho Bracket Paste LC (Bisco), (**b**) Light-Cure Orthodontic Paste (Leone), and (**c**) Transbond XT^TM^ Light Cure Adhesive (TXT) surfaces.

**Table 1 materials-14-02485-t001:** Chemical composition of the light-cured orthodontic adhesive resins considered in the study.

Orthodontic Composite Resin	Manufacturer	Composition	Acronym
Bisco Ortho Bracket Paste LC	Bisco, Schaumburg, Illinois, USA	UDMA (5–10%), TEGDMA (5–10%), molten silicon (50–75%). The substances contained in the remaining part of the adhesive are not specified by the supplier	Bisco
Light-Cure Orthodontic Paste	Leone s.p.a., Sesto Fiorentino, FI, Italy	Bis-GMA, UDMA, TEGDMA, silica and other inert fillers, catalysts and stabilizers (unknown percentages since not provided by the manufacturer)	Leone
Transbond XT^TM^ Light Cure Adhesive	3M Unitek, Monrovia, CA, USA	Bis-GMA (5–10%), bis-EMA (10–20%), TEGDMA (5–10%), reaction products with quartz (70–80%), reaction products with dichlorodimethylsinane with silica (<2%)	TXT

**Table 2 materials-14-02485-t002:** Average roughness (R_a_), root-mean-square roughness (R_q_), and average depth (R_z_) of the Bisco Ortho Bracket Paste LC (Bisco).

Bisco Measurements	R_a_ (µm)	R_q_ (µm)	R_z_ (µm)
1	0.425	0.601	3.885
2	0.408	0.487	2.643
3	0.396	0.587	4.500
4	0.209	0.309	2.307
5	0.577	0.826	4.459
6	0.261	0.327	1.656
7	0.757	0.911	4.151
8	0.802	1.134	7.300
9	0.563	0.715	3.543
10	0.880	1.095	4.656
11	0.751	1.009	4.600
12	0.442	0.556	2.463
**Average**	**0.539**	**0.713**	**3.847**
**Standard Deviation**	0.219	0.283	1.498

**Table 3 materials-14-02485-t003:** Average roughness (R_a_), root-mean-square roughness (R_q_), and average depth (R_z_) of the Light-Cure Orthodontic Paste (Leone).

Leone Measurements	R_a_ (µm)	R_q_ (µm)	R_z_ (µm)
1	0.315	0.414	2.256
2	0.219	0.292	1.233
3	0.488	0.656	3.391
4	0.403	0.635	4.178
5	0.723	1.081	5.611
6	0.263	0.329	1.842
7	0.622	0.814	4.634
8	0.815	1.264	7.173
9	0.500	0.716	4.265
10	0.727	0.856	3.566
11	0.390	0.481	2.449
12	0.945	1.081	5.611
**Average**	**0.534**	**0.718**	**3.851**
**Standard Deviation**	0.232	0.314	1.753

**Table 4 materials-14-02485-t004:** Average roughness (R_a_), root-mean-square roughness (R_q_), and average depth (R_z_) of the Transbond XT^TM^ Light Cure Adhesive (TXT).

TXT Measurements	R_a_ (µm)	R_q_ (µm)	R_z_ (µm)
1	0.326	0.473	2.293
2	0.392	0.508	3.015
3	0.601	0.821	3.410
4	0.495	0.663	3.240
5	0.722	0.947	4.873
6	0.335	0.430	2.234
7	0.323	0.420	2.616
8	0.346	0.421	2.215
9	0.591	0.716	3.455
10	0.451	0.561	2.529
11	0.260	0.479	5.217
12	0.816	1.029	6.175
**Average**	**0.472**	**0.622**	**3.439**
**Standard Deviation**	0.176	0.213	1.304

**Table 5 materials-14-02485-t005:** Absorbance values for the three examined resins in three different solution concentrations.

Material	UFC/mL 10^8^	UFC/mL 10^6^	UFC/mL 10^4^
BISCO	0.65	0.50	0.30
Leone	0.67	0.52	0.28
TXT	0.68	0.53	0.32

## Data Availability

Data available in a publicly accessible repository.

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
