# Peer review of "In Vitro Evaluation of Structural Factors Favouring Bacterial Adhesion on Orthodontic Adhesive Resins"

_materials, 2021, doi:10.3390/ma14102485_

Round 1

Reviewer 1 Report

The authors state about bacterial adhesion without really evaluating any in vitro studies with bacteria. They only conclude about the performance taking into consideration surface roughness and robustness. 

I do not think this paper should be accepted as more studies with different species of bacteria (adhesion, proliferation, live and dead...) or other parameters such as contact angle, proteins deposition, etc. should be complemented. 

Author Response

The authors have considered the comments of the reviewer and they have decided to further implement the findings highlighted in the manuscript with contact angle measurements and bacteria colony concentrations tests. The reviewer can find these additional experiments and their discussion in the revised paper and in the submitted file of the point-by-point response.

Reviewer 2 Report

Notes to the article:

L16-L26: 1. The abstract needs improvement. The authors wrote one long sentence. It is incomprehensible in this form. 

L90-L97: The purpose of the work is not clearly stated. It relates to research methods and not to a scientific problem. 

L117-L188: Please extend the explanation as to why sample finishing and polishing was not performed. How does this method of sample preparation translate into the clinical situation? 

L171 - L176: Why was the roughness parameter Rv not used?

L250-L375 (Discussion): The discussion did not clearly explain how the FIB analysis relates to the research problem. The nature of the research problem was explained to a greater extent on the basis of the work of other researchers than on the basis of the results of the research presented in the article. 

L400 - L404: This fragment can be moved to the discussion chapter. 

This fragment can be moved to the discussion chapter. Conclusions are not synthetic and contain new threads. It is a continuation of the discussion chapter. The most important conclusions should be included in the conclusions. Conclusions should relate to a greater extent to the tested materials. It is possible that the contact angle and surface free energy tests would be useful to achieve the research objective. 

Author Response

We thank the reviewer for these valuable advices. We improved the paper according to the reviewer comments. Please check the uploaded file for the point-by-point response.

Reviewer 3 Report

The manuscript described well the morphological features of three different resins. However, there are a few issues that need to be fixed.

  • The authors talked about differences related to chemical compositions: one has quartz and two have silica. Can the authors provide EDX analysis of the surfaces in order to verify the distribution of the fillers?it probably will help to verify the distribution and its possible influence on the roughness.
  • another test to verify the antibacterial properties could be the contact angle to see if the composition and the roughness influenced the wettability of the resins.
  • conclusions are to long. Can the author make it shorter?

Author Response

The authors thank the reviewer for the comments and the valuable advices for improving the manuscript. We added contact angle measurements and bacterial colony concentrations to confirm that the evidences observed in the surface roughness of the materials can indicate the potential affinity of resins in bacterial adhesion. Please check the revised version of the paper and the point-by-point response attached here.
